# Microbiological diagnosis and mortality of tuberculosis meningitis: Systematic review and meta-analysis

**Getachew Seid**[1,2]*, **Ayinalem Alemu**[1,2], **Biniyam Dagne**[1,2], **Dinka Fekadu Gamtesa**[1]

**1** Ethiopian Public Health Institute, Addis Ababa, Ethiopia, **2** Aklilu Lemma Institute of Pathobiology, Addis Ababa University, Addis Ababa, Ethiopia

* gech1365@gmail.com

**Data Availability Statement:** All relevant data are within the paper and its Supporting information files.

# Abstract

## Background

Tuberculosis (TB) which is caused by *Mycobacterium tuberculosis* poses a significant public health global treat. Tuberculosis meningitis (TBM) accounts for approximately 1% of all active TB cases. The diagnosis of Tuberculosis meningitis is notably difficult due to its rapid onset, nonspecific symptoms, and the difficulty of detecting *Mycobacterium tuberculosis* in cerebrospinal fluid (CSF). In 2019, 78,200 adults died of TB meningitis. This study aimed to assess the microbiological diagnosis TB meningitis using CSF and estimated the risk of death from TBM.

## Methods

Relevant electronic databases and gray literature sources were searched for studies that reported presumed TBM patients. The quality of included studies was assessed using the Joanna Briggs Institute Critical Appraisal tools designed for prevalence studies. Data were summarized using Microsoft excel ver 16. The proportion of culture confirmed TBM, prevalence of drug resistance and risk of death were calculated using the random-effect model. Stata version 16.0 was used perform the statistical analysis. Moreover, subgroup analysis was conducted.

## Results

After systematic searching and quality assessment, 31 studies were included in the final analysis. Ninety percent of the included studies were retrospective studies in design. The overall pooled estimates of CSF culture positive TBM was 29.72% (95% CI; 21.42–38.02). The pooled prevalence of MDR-TB among culture positive TBM cases was 5.19% (95% CI; 3.12–7.25). While, the proportion of INH mono-resistance was 9.37% (95% CI; 7.03–11.71). The pooled estimate of case fatality rate among confirmed TBM cases was 20.42% (95%CI; 14.81–26.03). Based on sub group analysis, the pooled case fatality rate among HIV positive and HIV negative TBM individuals was 53.39% (95%CI; 40.55–66.24) and 21.65% (95%CI;4.27–39.03) respectively.

**Funding:** The author(s) received no specific funding for this work.

**Competing interests:** The authors have declared that no competing interests exist.

## Conclusion

Definite diagnosis of TBM still remains global treat. Microbiological confirmation of TBM is not always achievable. Early microbiological confirmation of TBM has great importance to reduce mortality. There was high rate of MDR-TB among confirmed TBM patients. All TB meningitis isolates should be cultured and drug susceptibility tested using standard techniques.

## Introduction

Tuberculosis(TB) poses a significant public health global threat, which is caused by *Mycobacterium tuberculosis*(Mtb) bacteria. According to the World Health Organization (WHO), in 2020, the number of people newly diagnosed with TB dropped to 5.8 million with 1.3 million TB deaths among HIV-negative people and an additional 214 000 among HIV-positive people [1]. Following a primary or post-primary pulmonary infection, *Mycobacterium tuberculosis* can attack any part of the body including the central nervous system. Tuberculosis meningitis (TBM) is the most common type of central nervous system TB. Some patients who have or have had tuberculosis may develop the rare complication known as tuberculous meningitis. Tuberculous meningitis accounts for approximately 1% of all cases of active tuberculosis [2].

Southeast Asia and Africa accounted for 70% of global TBM incidence. WHO estimated that 78,200 (95% UI; 52,300–104,000) adults died of TBM in 2019. Tuberculous Meningitis case fatality in those treated was on average 27% [3, 4]. Besides, TBM can cause a diverse clinical picture including altered mental status, meningitic features, seizures, cranial nerve palsies, and focal neurological deficits [5]. It is among severe diseases which account 5–10% of extrapulmonary tuberculosis cases [2].

The disease involves the infection of the meninges of the host, which is caused by Mtb and other mycobacteria. Over half of TBM survivors have neurological disability [6]. Patients with TBM usually required admission to the intensive care unit. The most predisposed populations to develop TBM are children under four years, the elderly and HIV-positive patients [7]. The challenge TBM management concentrated on rapid reliable diagnosis andtreatment. Drug resistance and HIV infection increase the difficulty of TBM management [8].

Following TB infection infants have an up to 20% risk of developing TBM. Over half of all children with tuberculosis in the world go undiagnosed or unreported. Tuberculous meningitis mostly develops within 2–6 months following primary pulmonary infections during childhood [9]. To diagnose TBM in children MRI is superior to CT imaging but its high cost and need for infrastructure make difficult to use it [10]. In children, Most of the time TBM presents as subacute meningitis which makes it difficult to distinguishes from other meningoencephalitis diseases [11].

The diagnosis of tuberculous meningitis is notably difficult due to its rapid onset, nonspecific symptom, and the difficulty of detecting *Mycobacterium tuberculosis* in cerebrospinal fluid (CSF) [12]. The examination of the cerebrospinal fluid is the gold standard for diagnosing TBM. The identification of tuberculous bacilli in the CSF, either by smear examination or by culture, is required for a definitive diagnosis [13]. Even though culture is the gold standard for diagnosing *Mycobacterium tuberculosis*, long time for Mycobacterium growth on Mycobacterium growth indicator tube (MGIT) and LJ medium may lead to a delay in diagnosis [14].

Tuberculosis meningitis diagnosis is challenging by several factors, particularly in low- and middle-income countries: first, CSF collection necessitates lumbar puncture; second, CSF

processing necessitates adequate laboratory capacity; and finally, available laboratory diagnosis methods (smear microscopy, molecular tests such as Xpert MTB/RIF, or CSF culture) have moderate sensitivity [15]. A lumbar puncture is performed by a doctor who is specially trained to collect CSF. In a diagnostic Lumbar Puncture, standard bedside aseptic procedures apply with no-touch technique [15]. At this time there were obstacles in the diagnosis of TBM due to the absence of quick, reliable and affordable diagnostic tests. This study aims to assess the microbiological diagnosis of TBM using CSF and to estimate case fatality rate from TBM.

## Methods

### Protocol and registration

The protocol of this systematic review and meta-analysis was registered on the PROSPERO (International Prospective Register of Systematic Reviews), University of York. It was assigned a registration number CRD42022323629.

### Literature search

Systematic literature searching was performed using the PubMed, EMBASE databases and gray literature to assess microbiological diagnosis and mortality of Tuberculosis meningitis. The Preferred Reporting Items for Systematic Reviews and Meta-Analysis (PRISMA) checklist [16] was used to conduct this systematic review and meta-analysis (S1 Table). There was no need for ethical approval because this study was based on previously published primary investigations. The following key terms were used to extract the intended data: Tuberculosis, meningitis, Tuberculous meningitis, diagnosis, microbiological diagnosis bacteriologically confirmed, mortality, fatality, death and TB culture.

The search terms and their variations were used in combination. The Boolean operators AND and OR were used accordingly. Articles were limited to papers published in the English language without a limit of a published year. The final search included studies published up to May 1, 2022.

### Selection criteria

Included studies were: (1) original study on TBM presumptive patients; (2) published in the English language without regard to a publication year; 3). having described microbiological diagnosis of tuberculous meningitis based on CSF Mycobacteriological culture result data. Additionally, included articles should be peer-reviewed, fulfilled the above listed inclusion criteria and adequately addresses the objective of the study. Studies with incomplete data, studies not used culture technique to diagnose TBM, and review articles, meta-analyses and duplicates were all excluded from the study. Two authors (GS and AA) search and selected articles based on their title and abstract. Additionally, they did independent screening of the full text of the retrieved article to be included in the final analysis.

### Data extraction

To collect pertinent data from each eligible study, a pre-designed Microsoft 2010 excel data extraction form was used. The extraction activity was carried out by two writers (GS and BD). The quality and completeness of the extracted data were also reviewed by the third Author (DF). The following information was extracted: initial author name; year of publication; country of study, study period, age of study participants; study design, sample size of participants, case fatality rate, MDR-TB prevalence, and INH mono-resistance prevalence.

## Quality assessment

The Joanna Briggs Institute Critical Appraisal (JBI) techniques for prevalence studies were used to assess the quality of eligible papers [17]. There are nine quality indicators on the JBI checklist for the prevalence study. These quality indicators were converted to 100%, and the quality score was assessed as high if >80%, medium if 60–80%, and low if <60%. Two authors (GS and BD) carried out the quality assessment, while the third author handled the disagreement between the two authors (AA).

## Data analysis

Data were summarized and saved in Microsoft Excel 2016 before being exported to STATA Version 16.0 for analysis. All studies were pooled to estimate the risk of death of Tuberculosis meningitis presumptive patients at any age. Subgroup analysis was done based on the age of study participants (children or adult), HIV status and study design. Heterogeneity among studies was examined using forest plots and $I^2$ heterogeneity tests. In the current review, $I^2 > 50\%$ a random effect model was used for analysis. Funnel plot and an Egger's test (p-value 0.1 as a significant level) to see if there was any potential for publication bias. The forest plot provides a visual inspection of the confidence intervals of effect sizes of individual studies. The presence of non-overlapping intervals suggests heterogeneity.

# Result

## Eligible studies

Using the study's search terms, 1354 studies were found through a systematic search of electronic databases. After removing 1122 duplicate research, titles and abstracts were used to screen 232 publications. 174 studies were removed from the full-text review based on the abstract and title review. Only 31 [18–48] papers were included in the final systematic review and meta-analysis after full-text review of 54 studies (Fig 1).

## Study characteristics

There were 14 studies from Asia, eight from Europe, five from America, and only four [20, 26, 27, 36] studies from Africa (3 in South Africa and one in Uganda). Ninety percent of the included studies were retrospective studies in design. The study period of the studies was from 1985 to 2020. The range of sample sizes was 20 [23] to 6762 [36] study participants. Five studies [18, 20, 25, 27, 32] were conducted on children under the age of 18 and seven studies were conducted on adults over the age of 18. The rest studies included all study participants without discrimination on age. The total study participants of the included studies were 20,596 (Table 1).

Quality assessments of the included studies are provided in the (S2 Table). Ten studies [19, 21, 22, 23, 28, 30, 33, 34, 38, 47] score medium quality based on JBI quality assessment checklist for prevalence studies. While most of the studies score high quality using JBI checklist for prevalence studies.

## Microbiological diagnosis

The overall pooled estimate of Tuberculosis meningitis confirmed by CSF culture was 29.72% (95% CI; 21.42–38.02). The lowest percentage of TBM confirmed by CSF culture was 1.64% [22] and the highest percentage was 85.13% [34] (Fig 2). Prevalence of definite TBM diagnosed by AFB microscopy was 10.04% (95% CI; 4.31–15.78) (Fig 3).

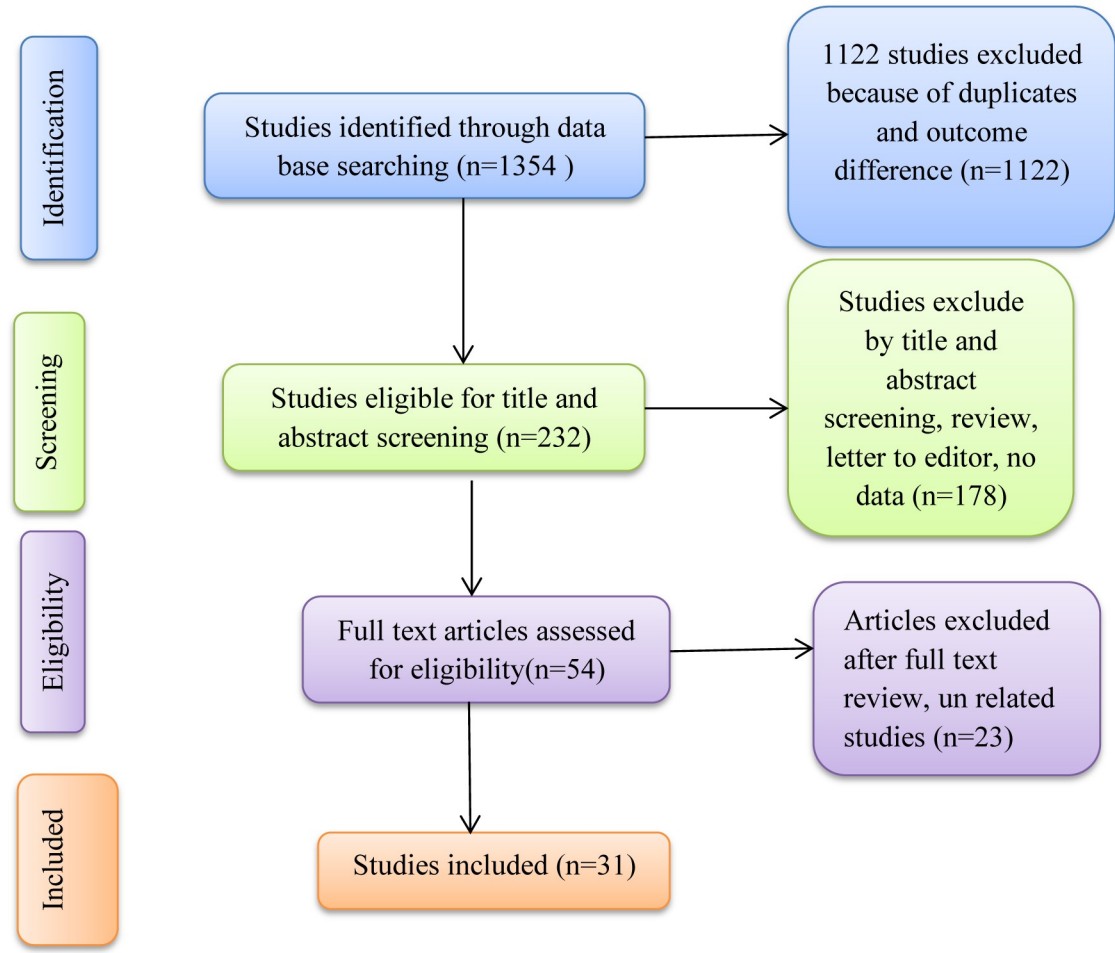

**Fig 1. Flow diagram of systematic search of studies for this systematic review and meta-analysis.**

Only fourteen studies reported the drug resistance pattern of the CSF culture-positive isolates. A total of 2736 CSF Mycobacterium TB culture-positive isolates were tested for drug susceptibility. Fourteen studies(5 from india,4 from china,2 from south Africa,1 from America,1 from Peru and 1 from Vietnam) were included to analyses the drug resistance pattern. MDR-TBM was found in 5.19% of these isolates (95% CI: 3.12–7.25) (Fig 4). Eight studies reported the proportion of INH mono resistance from the above total isolates. INH mono-resistance was 9.37% (95% CI; 7.03–11.71) (Fig 5).

## Case fatality rate among TBM patients

The proportion of TBM patients who died was reported in twenty-one studies. There were 1250 deaths out of a total of 6896 TBM patients. The estimated case fatality rate in TBM patients was 20.42% (95%CI; 14.81–26.03) (Fig 6).

## Sub-group analysis of case fatality among TBM patients

A subgroup analysis of case fatality rates by age, study design type, and HIV status yields estimates of 9.80% (95% CI;3.22–16.37) in children under the age of 18 and 24.82% (95% CI;17.05–32.59) in adults (greater than or equal to 18 years old); 20.34% (95% CI;14.03–26.65)

**Table 1. Study characteristic of included studies.**

| Author_year | Country | Study period | Study design | Participant age | Sample size |
|---|---|---|---|---|---|
| Ali, et al. 2015 [18] | Diyarbakir Turkey | 1998 to 2008 | Retrospective | <18 | 185 TBM |
| Anne-Sophie, et al. 2011 [19] | Denmark | January 2000 to December 2008 | Retrospective | All age | 50 TBM |
| Anu, et al. 2018 [20] | S/Africa | 2010–2014 | Retrospective | 3 months-15 years | 865 TBM |
| Baobao, et al. 2021 [21] | Shandong, China | January 2008 to April 2018. | Retrospective | >18 | 80 TBM |
| Chia, et al. 2017 [22] | Kebangsaan Malaysia | January 2003 to February 2015 | Observational | >18 | 61 TBM |
| Christiene, et al. 2002 [23] | Denmark | 1988 to July 2000. | Retrospective | All age | 20 TBM |
| Cíntia Helena, et al. 2014 [24] | Brazil | 2001 to 2010 | Descriptive | All age | 116 TBM |
| Dong-Mei, et al. 2020 [25] | Southwest of China | January 2013 to December 2018 | Retrospective | < 14 years old | 319 TBM |
| Fiona, et al. 2020 [26] | Uganda | Nov 25, 2016, to Jan 24, 2019 | Retrospective | >18 | 204TBM |
| Gijs, et al. 2009 [27] | South Africa | January 1985 to April 2005 | Retrospective | <18 | 554TBM |
| Heng, et al. 2016 [28] | Sabah, Malaysia | February 2012 to March 2013 | cohort | >12 | 84 TBM |
| Hosoglu, et al. 2003 [29] | Turkey | 1985 to 1998 | Retrospective | >18 | 469TBM |
| Jaime, et al. 2019 [30] | Peru | 2006 to 2015 | Retrospective | >18 | 263TBM |
| Renu, et al. 2017 [31] | India | July 2012 to July 2015 | Prospective | All age | 197 TBM |
| Robindra, et al. 2020 [32] | Europe | February 2016 to August 2016 | Retrospective | 0–16 years | 118 TBM |
| Yahia, et al. 2014 [33] | Qatar | January 2006 to December 2012 | Retrospective | >18 | 80 TBM |
| Christopher, et al. 2010 [34] | USA | 1 January 1993 to 31 December 2005 | Retrospective | All age | 1896TBM |
| Krishnapriya, et al. 2020 [35] | South India | August 2018 to February 2020 | Observational | | 293 TBM |
| Patel, et al. 2004 [36] | S/Africa | 1999 through 2002 | Retrospective | All age | 6762TBM |
| Ting, et al. 2016 [37] | Shaanxi, China | September 2010 to December 2012 | Retrospective | All age | 350 TBM |
| Jingya, et al. 2016 [38] | southwest China | - | - | 11 to 84 | 401 TBM |
| Kavitha, et al. 2016 [39] | India | May 2013 –April 2014 | Prospective | 3 months to 70 years | 698 TBM |
| Duc T, et al. 2019 [40] | America | 01/2010 to 12/2017 | Retrospective | All age | 192 TBM |
| Egidia, et al. 2015 [41] | Romania | 2004 to 2013 | Retrospective | All age | 204 TBM |
| Erdem, et al. 2013 [42] | Multi-country | 2000 to 2012. | Retrospective | All age | 506 TBM |
| Filiz, et al. 2011 [43] | Turkey | 1998 to 2009 | Retrospective | >14 | 160 TBM |
| Jyothi, et al. 2017 [44] | India | 2009 to 2014 | Retrospective | All age | 790 TBM |
| Lidya, et al. 2018 [45] | Indonesia | 2006 to 2016 | Cohort | >18 | 1180 TBM |
| Miguel, et al. 2020 [46] | Mexico | January 2015 to March2018 | Retrospective | ≥18 | 41 TBM |
| Nguyen, et al. 2014 [47] | Vietnam | 17 April 2011 to 31 December 2012 | Retrospective | >18 | 379 TBM |
| Syed, et al. 2017 [48] | India | 2013 to 2015 | - | >18 | 267 TBM |

and 30.92% (95% CI;18.40–43.44) in retrospective and other study designs, respectively; 53.39 (95%CI;40.55–66.24) in HIV positive TBM patients and 21.65 (95%CI;4.27–39.03) among HIV negative TBM patients (Table 2).

## Discussion

In this systematic review and meta-analysis the microbiological diagnosis of Tuberculosis meningitis and the risk of death among patients were calculated. According to the data around one–third of TBM patients had CSF microbiological (TB culture and AFB microscopy) confirmed illness. MDR-TB was shownto be prevalent in TBM patients. The risk of death was significant among TB meningitis patients. As per the findings, one patient will die for every five TBM cases.

The culture confirmed diagnostic rate reported in this study (29.72%) was slightly near to the report (38.9%) of a previous study [49]. It implies that 75% of TBM patients received anti-TB treatment empirically. This finding was also in support with the reports of previous study

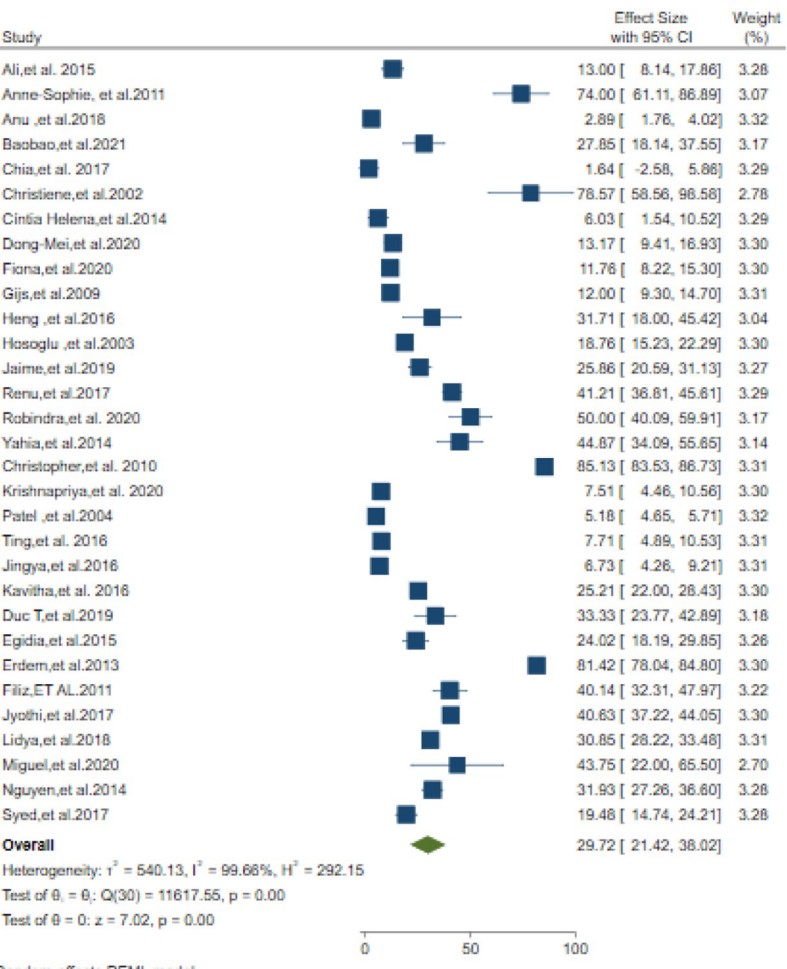

**Fig 2. CSF Culture confirmed Tuberculosis meningitis among suspected patients.**

which stated as in more than 50 per cent TBM patients, microbiological confirmation is not achieved This data indicated that conventional microbiological diagnosis of TBM tests has suboptimal positivity from CSF samples. Due to constrain of infrastructure and trained personnel, Worldwide there was a difficulty in diagnosing TBM using CSF. Junior doctors possess uncertainties regarding performing the procedure and frequently perform below expectations [50]. Lumbar puncture (LP) is often not performed in sub-Saharan African and other resource-limited settings [51]. Culture for *M. tuberculosis* performed on CSF had even lower positivity, producing a positive result in only approximately one in three cases [52].

Besides its longer turnaround time and inaccessibity, the lower positivity rate of CSF culture makes doubt its use as a gold standard diagnosis method for TBM. The positive rate of detection for the smear and culture tests is low alerting the globe to invest in rapid accurate and accessible diagnostic methods. Paucibacillarity of TBM makes it difficult to isolate *Mtb* in CSF by conventional culture methods. Even though rapid, sensitive and highly specific molecular detection methods have been favored, their cost and accessibility make early diagnosis of TBM difficult [53].

The lower positivity of CSF for *Mycobacterium tuberculosis* based on AF smear microscopy found in this meta-analysis was similar to other studies report which describe staining of CSF

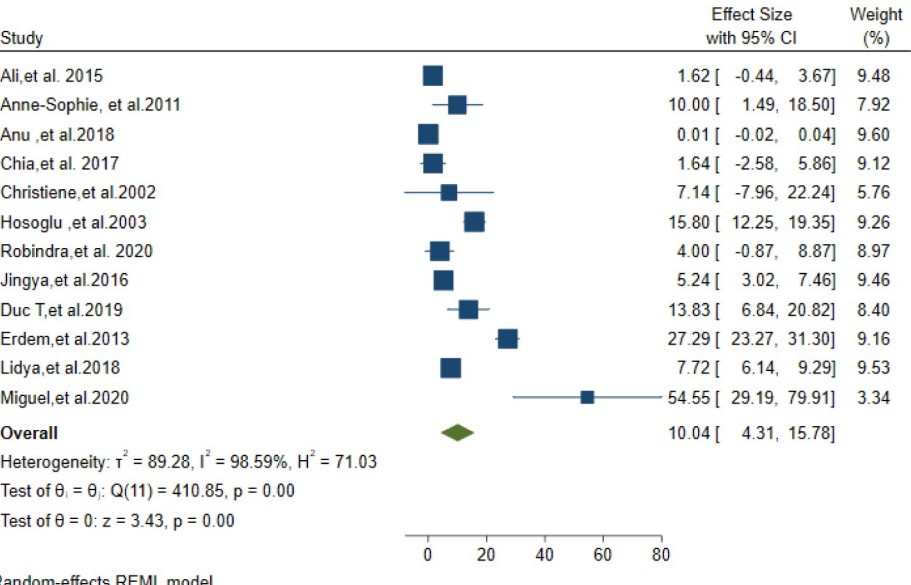

**Fig 3. ZN AFB microscopy positivity of CSF in TBM suspected patients.**

smears for acid-fast bacilli has poor sensitivity (about 10% to 15%) [54]. However, smear microscopy is the most widely used rapid and inexpensive diagnostic test for TB, especially in low and middle-income countries. Based on this most TBM cases were not microbiologically confirmed.

This systematic review and meta-analysis study has shown that drug resistance in TBM is not an unusual occasion. The rate of MDR-TB and INH mono resistance was 5.19% and

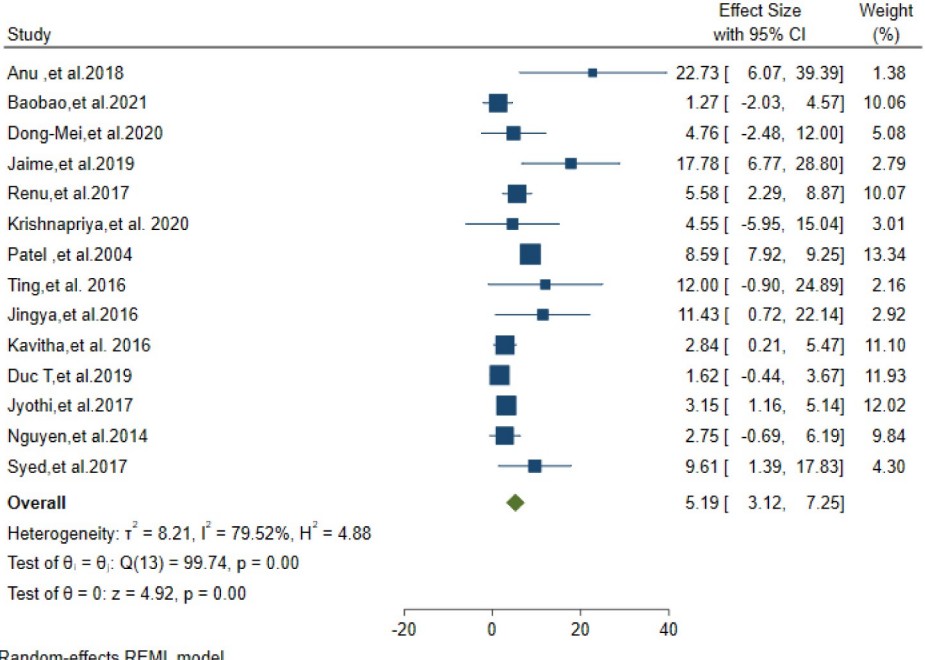

**Fig 4. Pooled estimate of MDR-TB prevalence in Tuberculosis confirmed isolates.**

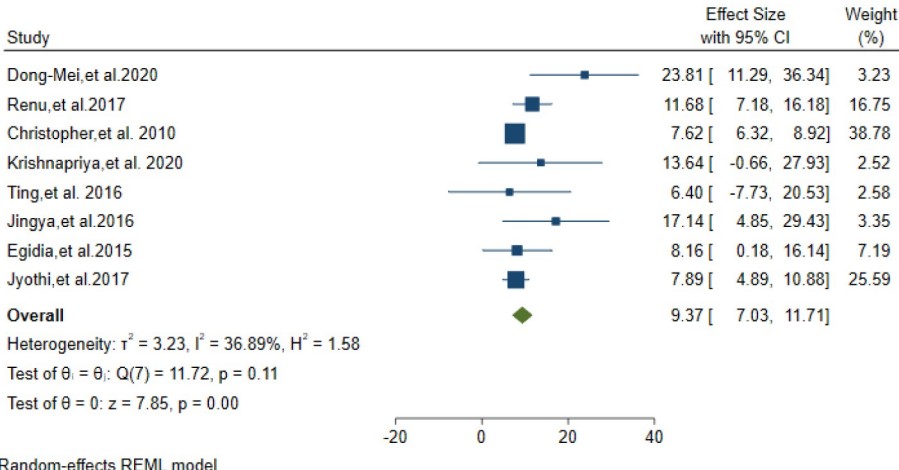

**Fig 5. Prevalence of INH mono resistance in Tuberculosis meningitis confirmed isolates.**

9.37% respectively. Since most of the included studies to analyze drug resistance pattern were from Asia (5 from India, 4 from china and 1 from Vietnam), the result reflects drug resistance pattern in that specific region. This indicates that TBM has a high vulnerability to drug resistance. Thus with the difficulties of getting precious CFS samples from TBM presumptive patients countries must include microbiological diagnosis of *Mycobacterium tuberculosis* in their national strategic plan and algorithm.

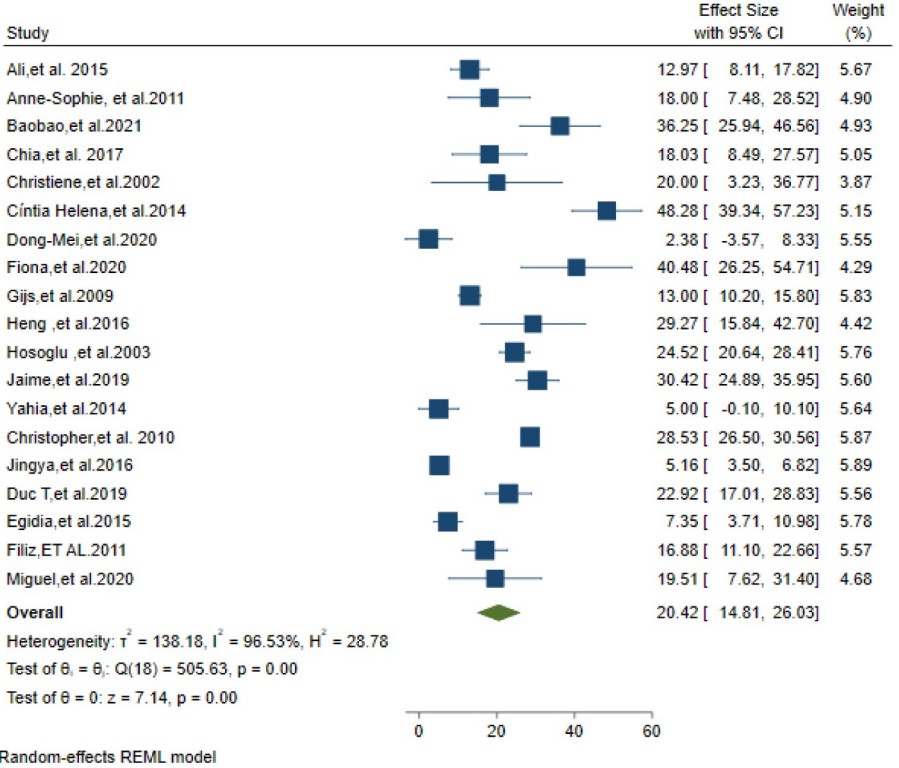

**Fig 6. Mortality among Tuberculosis meningitis suspected patients.**

**Table 2. Sub group analysis of mortality.**

| Characteristic | Number of studies | Number of deaths | Proportion of death (95%CI) |
|---|---|---|---|
| **Age** | | | |
| <18 years | 3 | 95 | 9.80 (3.22–16.37) |
| ≥18 years | 7 | 277 | 24.82 (17.05–32.59) |
| **Study type** | | | |
| Retrospective | 17 | 1076 | 20.34 (14.03–26.65) |
| other study design | 4 | 160 | 30.92(18.40–43.44) |
| **HIV status** | | | |
| Positive* | 4 | 220 | 53.39 (40.55–66.24) |
| Negative* | 4 | 173 | 21.65 (4.27–39.03) |

Note: *primary studies conducted mortality rate among HIV positive were Jaime, et al.2019 [30]; Christopher, et al. 2010 [34]; Cecchini, et al.2009 and Fiona, et al. 2020 [26].

*Primary studies conducted mortality rate among HIV negative were: Jaime, et al.2019 [30]; Christopher, et al. 2010 [34]; Cecchini, et al.2009 and Jingya, et al. 2016 [38]

According to the findings, 20.03% of TBM patients died during the course of their illness. It was alligned with the study finding of another study [55]. Our sub-group analysis showed that the risk of death was higher among adults (≥18 years) and HIV positive than their respective children (<18 years old) and HIV negative patients. Majority of the included studies were done after the initiation of antiretroviral treatment in most of developed and developing countries. The different case fatality rate reported in this study among children and adults was different from the reports of a previous single study [41] which found a similar 7.03% case fatality rate in both groups. This finding (mortality rate among children 9.8%) is lower than the report of previous systematic review and meta-analysis [56]. which reported 19.3% mortality rate among children. It might be due to the previous study participants were HIV–infected children. Among adults, our study finding was consistent with the previous studies [49, 55].

According to this study, HIV-TBM co-infected individuals have a two-fold greater case fatality rate than HIV-negative patients; mortality in HIV-negative TBM patients was 21.65%, compared to 53.39 percent in HIV-positive TBM patients. A prior study [49] found a mortality rate of 53.4 percent among adult HIV-positive TBM patients, which was similar to this. The HIV-infected person is at higher risk of developing disseminated extrapulmonary tuberculosis including TBM, particularly at a stage of more advanced immunosuppression [56]. It has been reported that tuberculosis patients co-infected with HIV were more likely to have poor treatment outcomes and death [57, 58].

There was a lot of heterogeneity between studies. We were able to find subgroup analysis based on the features of the included research, but we still don't know what caused the heterogeneity. Although we were unable to pinpoint the source of heterogeneity, the following factors could contribute to publication bias and heterogeneity: 1). We only considered research that was published in English; 2).the smallest sample size of the included studies was 20; and 3).the majority of the studies were retrospective.

Our study has some limitations: First, in this meta-analysis, we only included studies published in English. Second, we are unable to analyze case fatality by anti-retroviral therapy use and CD4 count due to a lack of sufficient data. Third, since, there was high heterogeneity of studies interpretation of results need attention.

## Conclusion

Tuberculosis meningitis cannot always be confirmed microbiologically. There was high rate of mortality in tuberculosis meningitis patients. The importance of early microbiological confirmation of TBM in reducing mortality is enormous. TBM patients have a high prevalence of MDR-TB infection. Tuberculous meningitis should be diagnosed using rapid, sensitive, and specific molecular testing methods. All TB meningitis isolates should be cultured and drug susceptibility tested using standard techniques. To investigate this goal in greater depth, prospective studies with a bigger sample size were required.

## Supporting information

**S1 Table. PRISMA checklist for systematic review and meta-analysis.**
(DOCX)

**S2 Table. Quality assessment of included studies.**
(DOCX)

**S3 Table. Raw data for the analysis.**
(XLSX)

## Acknowledgments

Our great acknowledge goes to the author of primary studies included in this systematic review and meta-analyses.

## Author Contributions

**Conceptualization:** Getachew Seid.

**Data curation:** Getachew Seid, Ayinalem Alemu, Biniyam Dagne.

**Formal analysis:** Biniyam Dagne, Dinka Fekadu Gamtesa.

**Methodology:** Dinka Fekadu Gamtesa.

**Supervision:** Ayinalem Alemu, Dinka Fekadu Gamtesa.

**Writing – original draft:** Getachew Seid.

**Writing – review & editing:** Getachew Seid, Ayinalem Alemu, Biniyam Dagne, Dinka Fekadu Gamtesa.

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
