## [Decision Letter · Decision Letter 0]

22 Aug 2022

PONE-D-22-14016Microbiological Diagnosis and Mortality of Tuberculosis Meningitis: Systematic Review and Meta-analysisPLOS ONE

Dear Dr. Abegaz,

Thank you for submitting your manuscript to PLOS ONE. After careful consideration, we feel that it has merit but does not fully meet PLOS ONE’s publication criteria as it currently stands. Therefore, we invite you to submit a revised version of the manuscript that addresses the points raised during the review process.

Please pay very careful attention to the comments provided below and make the requested changes

We look forward to receiving your revised manuscript.

Kind regards,

Muhammad Osman, MBChB, MSc, PhD

Academic Editor

PLOS ONE

Journal Requirements:

Additional Editor Comments:

Thank you for the submission of this important review.

Your manuscript requires a major revision and I request that you carefully consider the comments from the reviewers.

In addition, please note the following comments:

Lines 129-131 and Figure 1: 1354 studies identified with 147 duplicates removed, this should result in 1207 but the next block is 232

please correct the figure and text to document the exclusions of 975/1207 - with the reasons for exclusions

Figure 2 - uses a column heading effect size but this is actually the proportion not effect size (same in Fig 3, 4, 5, and 6)

Once you have addressed all the comments, please carefully review the syntax and grammar of your manuscript.

Reviewers' comments:

Reviewer's Responses to Questions

**Comments to the Author**

1. Is the manuscript technically sound, and do the data support the conclusions?

Reviewer #1: Partly

Reviewer #2: Yes

2. Has the statistical analysis been performed appropriately and rigorously? 

Reviewer #1: I Don't Know

Reviewer #2: Yes

3. Have the authors made all data underlying the findings in their manuscript fully available?

Reviewer #1: Yes

Reviewer #2: Yes

4. Is the manuscript presented in an intelligible fashion and written in standard English?

Reviewer #1: No

Reviewer #2: Yes

5. Review Comments to the Author

Reviewer #1: I have reviewed the article by Seid and colleagues on the “Microbiological diagnosis and Mortality of Tuberculosis Meningitis: Systematic Review and Meta-analysis

The overall pooled estimate of CSF culture positive Tuberculosis Meningitis (TBM) was 29.72% and the pooled estimate of mortality rate was 20.42%.

It is a topic of great clinical interest but several issues arise:

1. There is need for an extensive review of the grammatical syntax and spellings

2. The section on eligibility criteria should be rewritten for clarity

3. Inaccuracies. It is not conceivable that TBM occurs in people who have had tuberculosis. TBM is active tuberculosis (line 38).

4. Lines 57 and 59 are contradictory regarding the clinical presentation of TBM- the former suggests the presentation is rapid while the latter suggests a sub-acute presentation

5. The study case definition is not clear (line 93 – 94)

6. In lines 225 and 226 one expects lists but these are not presented

7. The high mortality in HIV seropositive participants was noted. Was this stratified by CD4 count or antiretroviral therapy use?

8. What is the role of molecular diagnostics like Xpert Ultra (Shen, 2021) and metagenomic sequencing in the diagnosis of TBM?

9. Some elements in the conclusion are not supported by the study results

Reviewer #2: Introduction

Line 67. is the lumbar puncture technique a challenge for clinicians in low= and middle- income countries? As in line 182 & 183.

Methods

Literature Search

Line 80. Why was leterature search limited to Pubmed, EMBASE and gray literature only?

Lines 85-87. Good key words.

Selection criteria

Lines 96-99. Only inclusion criteria is explicitly stated here. Exclusion criteria should also be clearly stated here and not just later on e.g. lines 229 -231.

Quality Assessment

Line 114. ...high if >80%, medium if 60-80%, and low if 80%. Is this supposed to read, "and low if <60%."

Results

Study characteristics

Line 138. What was the rational for excluding studies before 1985?

Sub-group analysis of mortality among TBM patients

Lines 163-168. Good description.

Discussion

Line 174. Could the high prevalence of MDR-TB be related to only part of the studies (14/310 reporting this fnding as in line 153?

Lines 178 & 179. Could empirical treatment as opposed to treatment guided by drug sensitivity testing be partly responsible for mortality in TBM?

6. PLOS authors have the option to publish the peer review history of their article (what does this mean?). If published, this will include your full peer review and any attached files.

Reviewer #1: No

Reviewer #2: No

---

## [Author Response · Author response to Decision Letter 0]

30 Aug 2022

Response to Reviewer

I. Editor comments

1. Lines 129-131 and Figure 1: 1354 studies identified with 147 duplicates removed, this should result in 1207 but the next block is 232

Yes this is clerical error that through the data bases search we found 1354 studies; 1122 duplicates removed and finally 232 studies undergoes title and abstract screening

2. Figure 2 - uses a column heading effect size but this is actually the proportion not effect size (same in Fig 3, 4, 5, and 6)

We accept this comment and really it is the proportion not effect size and corrected on the document

3. Once you have addressed all the comments, please carefully review the syntax and grammar of your manuscript.

We tried to correct the syntax and grammar of the manuscript through reading and editing

II. Reviewers' comments

Reviewer #1:

1. There is need for an extensive review of the grammatical syntax and spellings

We tried to correct the syntax and grammar of the manuscript through reading and editing

2. The section on eligibility criteria should be rewritten for clarity

The eligibility criteria for the study was rewritten as follows

To be included in this study articles must be: (1) original study on TBM suspected patients; (2) published in the English language without regard to a publication year; 3). having described microbiological diagnosis of tuberculous meningitis based on CSF Mycobacteriological culture result data.

3. Inaccuracies. It is not conceivable that TBM occurs in people who have had tuberculosis. TBM is active tuberculosis (line 38).

This sentence was rewritten as

Some patients who have or have had tuberculosis may develop the rare complication known as tuberculous meningitis. Tuberculous meningitis accounts for approximately 1 % of all cases of active tuberculosis

4.Lines 57 and 59 are contradictory regarding the clinical presentation of TBM- the former suggests the presentation is rapid while the latter suggests a sub-acute presentation

The reported duration of symptoms of tuberculous meningitis ranges between a single day and 6 months, presenting as acute, sub-acute, or chronic meningitis. In children, Most of the time TBM presents as sub-acute meningitis which makes it difficult to distinguishes from other meningoencephalitis diseases (11).

5. The study case definition is not clear (line 93 – 94)

We tried to rewrite the study case definition of microbiological diagnoses as follows.

Microbiological diagnoses of Tuberculosis meningitis confirms the suspicion of the disease and identify the etiologic agent by culture from CSF sample.

6. In lines 225 and 226 one expects lists but these are not presented

It is re-corrected in the main manuscript as follows

Although we were unable to pinpoint the source of heterogeneity, the following factors could contribute to publication bias and heterogeneity: 1).We only considered research that was published in English; 2).the smallest sample size of the included studies was 20; and 3).the majority of the studies were retrospective.

7. The high mortality in HIV seropositive participants was noted. Was this stratified by CD4 count or antiretroviral therapy use?

No this is not stratified by CD4 due to absence of enough data

8. What is the role of molecular diagnostics like Xpert Ultra (Shen, 2021) and metagenomic sequencing in the diagnosis of TBM?

As Xpert Ultra increases the sensitivity of Mycobacterium tuberculosis detection and its specificity is 100%, Xpert Ultra had an excellent diagnostic efficacy for TBM, and it could be the preferred initial test for TBM.(Shen,2021).Additionally it gives the result within 2 hours.

Even though metagenomic sequencing is not easily accessible for low and middle income countries, it has moderate sensitivity and high specificity for the diagnosis of tuberculous meningitis (Yu,2020).

9. Some elements in the conclusion are not supported by the study results

The conclusion part rewritten based on the findings of the study on the corrected manuscript.

Reviewer #2: Introduction

1.Line 67. Is the lumbar puncture technique a challenge for clinicians in low= and middle- income countries? As in line 182 & 183.

Yes in low and middle income country there is shortage of trained physicians to draw quality CSF sample through lumbar puncture. This affects the treatment and definite diagnosis of tuberculosis meningitis

2Methods

.Literature Search

Line 80. Why was literature search limited to Pubmed, EMBASE and gray literature only?

Because these were the major databases that will contain studies of our interest.Additionalyy in accessibility of other databases.

3.Lines 85-87. Good key words.

Selection criteria

Lines 96-99. Only inclusion criteria is explicitly stated here. Exclusion criteria should also be clearly stated here and not just later on e.g. lines 229 -231.

Studies with incomplete data, studies not used culture technique to diagnose TBM, and review articles, meta-analyses and duplicates were all excluded from the study.

3.Quality Assessment

Line 114. ...high if >80%, medium if 60-80%, and low if 80%. Is this supposed to read, "and low if <60%."

Really this is a clerical error and it is corrected on the revised manuscript

4.Results

Study characteristics

Line 138. What was the rational for excluding studies before 1985?

Here we did no exclude those studies published before 1985 rather the included studies for systematic review and meta-analysis were published from 1985 to 2020

5.Sub-group analysis of mortality among TBM patients

Lines 163-168. Good description.

 Thanks for your appreciation

6.Discussion

Line 174. Could the high prevalence of MDR-TB be related to only part of the studies (14/310 reporting this fnding as in line 153?

 Of 31 included studies only 14 report drug susceptibility result of the TBM isolates. There finding, pooled proportion of MDR-TB from these isolates is 5.19%,which support us to say there was high prevalence of MDR-TB among the confirmed tuberculosis meningitis patients.

7.Lines 178 & 179. Could empirical treatment as opposed to treatment guided by drug sensitivity testing be partly responsible for mortality in TBM?

 This could be supported by further detail trial study

---

## [Decision Letter · Decision Letter 1]

27 Sep 2022

PONE-D-22-14016R1Microbiological Diagnosis and Mortality of Tuberculosis Meningitis: Systematic Review and Meta-analysisPLOS ONE

Dear Dr. Abegaz,

Thank you for submitting your manuscript to PLOS ONE. After careful consideration, we feel that it has merit but does not fully meet PLOS ONE’s publication criteria as it currently stands. Therefore, we invite you to submit a revised version of the manuscript that addresses the points raised during the review process.

Please take note of the requested changes as detailed below.

We look forward to receiving your revised manuscript.

Kind regards,

Muhammad Osman, MBChB, MSc, PhD

Academic Editor

PLOS ONE

Journal Requirements:

Additional Editor Comments:

Dear Authors - Thank you for your response and edits to the manuscript.

Please take note of the following important changes:

line 10 - 'studies that reported culture confirmed TBM' this needs to be corrected - the aim was not to search for culture confirmed studies, the aim was to search for presumed/suspected TBM and then estimate the proportion that were culture confirmed

line 33 - 10 million cases of TB were detected - the WHO estimate is of burden (everyone who developed TB), detection is much lower and dropped further due to COVID disruptions

Line 68 - add some detail around the expertise and skill required to collect CSF by lumbar puncture (as in your previous response)

Line 94-95 the definition is not clear - what is this the definition for?

The discussion requires additional details on the limitations - you may either include these as you discuss the findings or expand the limitations paragraph. These should include:

- trained personnel - for what taking CSF, testing CSF, where is this limited? any publications that confirm this?

- rate of DR TB - this was only possible in subset of n=XX studies. Also check were these in a particular region and does this DR TB rate reflect underlying DR TB pattern in that region?

- HIV TBM mortality - study not able to stratify by CD4 count or ART use; also comment on the study period, could the high HIV TBM be because it occurred in period prior to widespread ART?

Figures - the heading effect size has not been changed as previously recommended and noted in the responses

Table 2: please annotate this table so that the HIV studies are specified. Add a footnote so that we can see which of the 4 studies were HIV+ and which were HIV- . The detail on age and study design is in table 1

Minor notes which reflect the need for copyediting and proof reading

the use of Mycobacterium tuberculosis (Mtb) - this needs consistency across the manuscript - please check nomenclature and adjust (lines 32, 35, 46, 61, 185, 195, 205 etc)

TBM to be used consistently

Incorrect capitalisation of words - line 53 Infants; line 72 Absence; line 185 Worldwide, line 197 Smear

Check tenses through out: line 108 'is' should be was

Language: line 156 'further analysis the drug resistance'; line 176 'than expected'; line 179 'definite diagnostic'; line 180 'were got'; line 181-182; line 189 'makes doubt its use';

Line 139 S/Africa needs to be spelled out; line 195 AF

line 139 continent could be deleted

line 143 'rest studies'

Reviewers' comments:

Reviewer's Responses to Questions

**Comments to the Author**

1. If the authors have adequately addressed your comments raised in a previous round of review and you feel that this manuscript is now acceptable for publication, you may indicate that here to bypass the “Comments to the Author” section, enter your conflict of interest statement in the “Confidential to Editor” section, and submit your "Accept" recommendation.

Reviewer #2: All comments have been addressed

2. Is the manuscript technically sound, and do the data support the conclusions?

Reviewer #2: Yes

3. Has the statistical analysis been performed appropriately and rigorously? 

Reviewer #2: Yes

4. Have the authors made all data underlying the findings in their manuscript fully available?

Reviewer #2: Yes

5. Is the manuscript presented in an intelligible fashion and written in standard English?

Reviewer #2: Yes

6. Review Comments to the Author

Reviewer #2: This is important work that needs further detailed studies to improve diagnosis and treatment outcome.

7. PLOS authors have the option to publish the peer review history of their article (what does this mean?). If published, this will include your full peer review and any attached files.

Reviewer #2: No

---

## [Author Response · Author response to Decision Letter 1]

12 Oct 2022

Point-by point response

1. Reference is corrected according to the journal guideline. Please see reference part page 10

2.line 10 - 'studies that reported culture confirmed TBM' this needs to be corrected - the aim was not to search for culture confirmed studies, the aim was to search for presumed/suspected TBM and then estimate the proportion that were culture confirmed

Yes the comment is correct and it is corrected as ‘..studies that reported presumed TBM patients ‘. See abstract part page 1.

3. Line 33 - 10 million cases of TB were detected - the WHO estimate is of burden (everyone who developed TB), detection is much lower and dropped further due to COVID disruptions

This editorial error is corrected as ‘…the number of people newly diagnosed with TB dropped to 5.8 million..’see page 1

4. Line 68 - add some detail around the expertise and skill required to collect CSF by lumbar puncture (as in your previous response)

Some of the skills and requirements for CSF collection are added to the manuscript .see page 2 line 72-74

5.Line 94-95 the definition is not clear - what is this the definition for?

Since the operational definition is still ambiguous we want to omit from the manuscript.

6.The discussion requires additional details on the limitations - you may either include these as you discuss the findings or expand the limitations paragraph. These should include:

- trained personnel - for what taking CSF, testing CSF, where is this limited? Any publications that confirm this?

In the discussion section we added some discussion on trained personnel and limited skill on taking CSF from lumbar puncture with references. See page 7 line 186-191

- Rate of DR TB - this was only possible in subset of n=XX studies. Also check was these in a particular region and does this DR TB rate reflect underlying DR TB pattern in that region?

The following sentence is added to the discussion part under rate of MDR-TB.’…Since most of the included studies to analyze drug resistance pattern were from Asia (5 from India, 4 from china and 1 from Vietnam), the result reflects drug resistance pattern in that specific region.

- HIV TBM mortality - study not able to stratify by CD4 count or ART use; also comment on the study period, could the high HIV TBM be because it occurred in period prior to widespread ART?

Majority of the included studies were done after the initiation of antiretroviral treatment in most of developed and developing countries. See page 8 line 221-223

7. Figures - the heading effect size has not been changed as previously recommended and noted in the responses

Sorry for the first response ,now it is corrected .please see each figures in the corresponding pages

8. Table 2: please annotate this table so that the HIV studies are specified. Add a footnote so that we can see which of the 4 studies were HIV+ and which were HIV- . The detail on age and study design is in table 1

Footnote is added to Table 2.please see page 21 for details.

9.Minor notes which reflect the need for copyediting and proof reading

- the use of Mycobacterium tuberculosis (Mtb) - this needs consistency across the manuscript - please check nomenclature and adjust (lines 32, 35, 46, 61, 185, 195, 205 etc)

All the nomenclatures were corrected according to the comment

-TBM to be used consistently

We make all the words consistent thought the document

Incorrect capitalization of words - line 53 Infants; line 72 Absence; line 185 Worldwide, line 197 Smear

Check tenses throughout: line 108 'is' should be was

Language: line 156 'further analysis the drug resistance'; line 176 'than expected'; line 179 'definite diagnostic'; line 180 'were got'; line 181-182; line 189 'makes doubt its use';

Line 139 S/Africa needs to be spelled out; line 195 AF

line 139 continent could be deleted

line 143 'rest studies'

Capitalization and grammar is corrected thought the document

---

## [Decision Letter · Decision Letter 2]

2 Dec 2022

Microbiological Diagnosis and Mortality of Tuberculosis Meningitis: Systematic Review and Meta-analysis

PONE-D-22-14016R2

Dear Dr. Abegaz,

We’re pleased to inform you that your manuscript has been judged scientifically suitable for publication and will be formally accepted for publication once it meets all outstanding technical requirements.

Kind regards,

Muhammad Osman, MBChB, MSc, PhD

Academic Editor

PLOS ONE

Additional Editor Comments (optional):

Dear Authors

Thank you for making the requested changes to the manuscript. In the current format, I have a concern that English language editing is required.

In addition to the points raised by reviewer 1 in their attached word document, please take note of the highlighted text and comments included in the attached pdf.

Reviewers' comments:

Reviewer's Responses to Questions

**Comments to the Author**

1. If the authors have adequately addressed your comments raised in a previous round of review and you feel that this manuscript is now acceptable for publication, you may indicate that here to bypass the “Comments to the Author” section, enter your conflict of interest statement in the “Confidential to Editor” section, and submit your "Accept" recommendation.

Reviewer #2: All comments have been addressed

2. Is the manuscript technically sound, and do the data support the conclusions?

Reviewer #2: Yes

3. Has the statistical analysis been performed appropriately and rigorously? 

Reviewer #2: Yes

4. Have the authors made all data underlying the findings in their manuscript fully available?

Reviewer #2: Yes

5. Is the manuscript presented in an intelligible fashion and written in standard English?

Reviewer #2: Yes

6. Review Comments to the Author

Reviewer #2: I believe this review adds something to the body of knowledge that we have. It has addressed some gaps while highlighting further areas for research.

7. PLOS authors have the option to publish the peer review history of their article (what does this mean?). If published, this will include your full peer review and any attached files.

Reviewer #2: No

---

## [Editor Report · Acceptance letter]

6 Feb 2023

PONE-D-22-14016R2 

Microbiological Diagnosis and Mortality of Tuberculosis Meningitis: Systematic Review and Meta-analysis 

Dear Dr. Seid:

I'm pleased to inform you that your manuscript has been deemed suitable for publication in PLOS ONE. Congratulations! Your manuscript is now with our production department. 

Kind regards, 

on behalf of

Dr. Muhammad Osman 

Academic Editor

PLOS ONE